# The Persistent, Pernicious Use of Pushbacks against Children and Adults in Search of Safety

Michael Garcia Bochenek [1,2]

1   Human Rights Watch, New York, NY 10118-3299, USA; michael.bochenek@columbia.edu
2   Institute for the Study of Human Rights, Columbia University, New York, NY 10027, USA

**Abstract:** Border pushbacks, including at the European Union's external borders and by countries such as Australia, Mexico, Turkey, and the United States, are common—and in fact have become a new normal. These border policing or other operations aim to prevent people from reaching, entering, or remaining in a territory. Screening for protection needs is summary or non-existent. Pushbacks violate the international prohibitions of collective expulsion and refoulement, and pushbacks of children are inconsistent with the best interests principle and other children's rights standards. Excessive force, other ill-treatment, family separation, and other rights violations may also accompany pushback operations. Despite formidable obstacles such as weak oversight mechanisms, undue judicial deference to the executive, and official ambivalence, domestic court rulings and other initiatives show some promise in securing compliance with international standards and affording a measure of accountability.

**Keywords:** asylum; migration; refugees; unaccompanied children; pushbacks; refoulement; collective expulsion; family separation; rule of law; European Court of Human Rights; Court of Justice of the EU

In November 2021, 16-year-old Farhad N. (not his real name), his younger sisters, and their parents were sheltering in a squat in northwest Bosnia and Herzegovina, 500 m from the Croatian border. They explained that they had fled their home in Afghanistan two years earlier, fearing for their lives as attacks by the Taliban escalated.

They had entered Croatia irregularly dozens of times, sometimes nearly reaching the capital, Zagreb. Each time, they said, Croatian authorities apprehended them, disregarded their requests for asylum, drove them to the border, and ordered them to wade across the river to Bosnia and Herzegovina.

Many of these pushbacks happened in the middle of the night. Sometimes Farhad and his family said they were able to walk directly into Velika Kladuša or another Bosnian town near the border, but often the Croatian police took them somewhere far from the regular border crossings. Because Croatian police usually seized and destroyed the family's phones, they explained they had no easy way of navigating to safety or calling for assistance.[1]

Pushbacks from Croatia to the non-E.U. countries it borders are common. Between January 2020 and December 2022, the Danish Refugee Council (DRC) recorded nearly 30,000 pushbacks from Croatia to Bosnia and Herzegovina. Approximately 13 percent of pushbacks recorded in 2022 were of children, alone or with families.[2] The office of the U.N. High Commissioner for Refugees and human rights groups have also recorded pushbacks from Croatia to Serbia.[3]

---

1   Interview with Farhad N. (pseudonym) in Una-Sana Canton, Bosnia (29 November 2021).
2   Danish Refugee Council (DRC), *Border Monitoring Factsheets* (March 2021–December 2022); DRC, *Bosnia and Herzegovina Border Monitoring Bimonthly Snapshot* (Jannuary/February 2021); DRC, *Border Monitoring Monthly Snapshots* (January–December 2020).
3   E.g., U.N. High Comm'r for Refugees (UNHCR), Statistical Snapshots: Serbia (August 2020–July 2022), https://www.unhcr.org/rs/en/country-reports (accessed on 14 February 2023); Border Violence Monitoring Network (2020, 2022); Human Rights Watch (2017).

Croatian pushbacks are often characterized by violence and deliberate humiliation. Video images captured by Lighthouse Reports, an investigative journalism group, for a 2021 investigation it conducted in collaboration with *Der Spiegel, The Guardian, Libération,* and other news outlets showed Croatian police in balaclavas forcing a group of people into Bosnia and Herzegovina. As *Der Spiegel* recounted, "One of the masked men repeatedly lashes out with his baton, letting it fly at the people's legs so that they stumble into the border river, where the water is chest-high. Finally, he raises his arm threateningly and shouts, 'Go! Go to Bosnia!'" (Christides (2021). See also Statius et al. (2021); Deeb et al. (2021).)

It Is not uncommon to hear accounts from men and teenage boys who have been made to walk back to Bosnia and Herzegovina barefoot and shirtless. Sometimes Croatian police have forced them to strip down to their underwear or, in a few cases, to remove their clothing completely. (See, e.g., Council of Europe Commissioner for Human Rights (2021, para. 16(vii)); Border Violence Monitoring Network (2020, 2022).) One group of men arrived at a refugee camp in Bosnia and Herzegovina with orange crosses spray-painted on their heads (Centre for Peace Studies and Welcome Initiative 2022; European Committee for the Prevention of Torture 2021a).

Teenage boys and adult men have regularly described facing beatings by Croatian police during pushbacks, sometimes as they are forced to run a gauntlet. Police have sometimes also shoved or struck women and younger children, witnesses explain, adding that younger children regularly see their fathers, older brothers, and other relatives punched, struck with batons, kicked, and shoved. Croatian border police have also discharged firearms close to children or pointed firearms at children. (See, e.g., Border Violence Monitoring Network (2022).)

Pushbacks are traumatizing for everybody (Marković et al. 2023). In addition, pushbacks increase the time everybody spends in transit. For children and their families, who frequently cannot travel as fast on foot as single adults can, pushbacks may add considerably to the time spent in difficult, often squalid, and potentially unsafe conditions before they are able to make a claim for asylum in an E.U. country. They increase the time children spend without access to formal schooling. For unaccompanied children in particular, pushbacks can increase the risk that they will be subject to trafficking (U.N. Special Rapporteur on Torture 2018).

Family separation may result from pushbacks: the nongovernmental organization Are You Syrious has reported cases of women allowed to seek asylum in Croatia with their children while their husbands are pushed back to Bosnia (E.U. Fundamental Rights Agency 2021b).

Pushbacks, broadly meaning "proactive operations aimed at physically preventing migrants from reaching, entering or remaining" (U.N. Special Rapporteur on Torture (2018, para. 49). See also U.N. Special Rapporteur on Human Rights of Migrants (2021, paras. 34–38); Council of Europe Commissioner for Human Rights (2021, pp. 7, 16).) in a territory and which employ summary screening for protection needs (or lack any screening at

all), violate the international prohibitions of collective expulsions[4] and refoulement,[5] the sending of people to places where they would face ill-treatment or other irreparable harm. Pushbacks of children are inconsistent with specific international standards that call on states to take particular care to ensure that returns of children are in their best interests.[6] Excessive force, other ill-treatment, family separation, and other rights violations may also accompany pushback operations.

This article assesses state practice—as documented in news reports, the findings of human rights and humanitarian groups, and investigations by domestic, regional, and international bodies—against these international norms. It includes a focus on children because children experience specific harms from pushback practices, whether they travel on their own or with family members; because the phenomenon of and issues faced by children in situations of migration are too often overlooked (Bhabha 2016; White et al. 2011; Nolan 2020); and because international children's rights norms offer particularly robust protections—in principle, if not in practice (Vaghri et al. 2019).

These sources of information are remarkably consistent. It is common to hear reports of pushbacks at the European Union's external borders. Other states that have regularly carried out pushbacks include Australia, Mexico, Turkey, and the United States.

Pushbacks have become, in short, a new normal.

The intention of pushbacks by states is primarily to avoid triggering the legal, administrative, and policy responsibilities that come with hosting an asylum seeker on the territory. Pushbacks serve effectively as a form of extra-legal border control.

---

4　Collective expulsions are prohibited under the International Covenant on Civil and Political Rights (ICCPR) and regional treaties, and the International Law Commission's special rapporteur on the expulsion of aliens suggested in 2007 that the prohibition of collective expulsions is a general principle of customary international law. See International Covenant on Civil and Political Rights, art. 13, 19 December 1966, 999 U.N.T.S. 171 [hereinafter "ICCPR"]; U.N. Human Rights Committee, General Comment No. 15: The Position of Aliens Under the Covenant, para. 10 (27th sess., 1986) ("article 13 would not be satisfied with laws or decisions providing for collective or mass expulsions"), in Compilation of General Comments and General Recommendations Adopted by Human Rights Treaty Bodies, at 18, 21, U.N. Doc. HRI/GEN/1/Rev.1 (29 July 1994); Protocol No. 4 to the Convention for the Protection of Human Rights and Fundamental Freedoms, Securing Certain Rights and Freedoms Other than Those Already Included in the Convention and the First Protocol Thereto, art. 4, 16 September 1963, E.T.S. No. 46 ("Collective expulsion of aliens is prohibited."); Charter of Fundamental Rights of the European Union, art. 19(1), O.J. C 326/391 (26 October 2012) ("Collective expulsions are prohibited."); American Convention on Human Rights, art. 22(9), 22 November 1969, 1144 U.N.T.S. 143 ("The collective expulsion of aliens is prohibited."); African Charter on Human and People's Rights, art. 12(5), 27 June 1981, 1520 U.N.T.S. 217 ("The mass expulsion of non-nationals shall be prohibited."); Int'l Law Comm'n, Third Report on the Expulsion of Aliens, para. 115, U.N. Doc. A/CN.4/581 (19 April 2007) ("it seems reasonable to suggest that there is a general principle of international law on this matter that is 'recognized by civilized nations' and prohibits collective expulsion").

5　The prohibition on refoulement is set forth in the Refugee Convention, the Convention against Torture, the ICCPR, the Convention on the Rights of the Child, the European Convention on Human Rights, and the E.U. Charter of Fundamental Rights, among other treaties, and is a norm of customary international law. Convention relating to the Status of Refugees, art. 33(1), 28 July 1951, 189 U.N.T.S. 137 [hereinafter Refugee Convention]; Convention against Torture and Other Cruel, Inhuman or Degrading Treatment or Punishment, art. 3, 10 December 1984, 1465 U.N.T.S. 85 [hereinafter Convention against Torture]; ICCPR, *supra* note 4, art. 7; Convention on the Rights of the Child, 20 November 1989, 1577 U.N.T.S. 3; Convention on the Protection of Human Rights and Fundamental Freedoms, 4 November 1950, E.T.S. No. 5 [hereinafter European Convention on Human Rights]; Charter of Fundamental Rights of the European Union, *supra* note 4, art. 19(2); Human Rights Comm., General Comment No. 20: Article 7 (Prohibition of Torture or Other Cruel, Inhuman or Degrading Treatment or Punishment), para. 9 (44th sess., 1992), in Compilation of General Comments, *supra* note 4, at 30, 31 ("States parties must not expose individuals to the danger of torture or cruel, inhuman or degrading treatment or punishment upon return to another country by way of their extradition, expulsion or refoulement."); Comm. on the Rights of the Child, General Comment No. 6: Treatment of Unaccompanied and Separated Children Outside Their Country of Origin, para. 27, U.N. Doc. CRC/GC/2005/6 (1 September 2005) ("States shall not return a child to a country where there are substantial grounds for believing that there is a real risk of irreparable harm to the child, such as, but by no means limited to, those contemplated under articles 6 and 37 of the Convention, either in the country to which removal is to be effected or in any country to which the child may subsequently be removed."); Lauterpacht and Bethlehem (2003, p. 139).

6　U.N. Comm. on Migrant Workers & U.N. Comm. on the Rights of the Child, Joint General Comment No. 3 (Comm. on Migrant Workers) and No. 22 (Comm. on the Rights of the Child) on the General Principles Regarding the Human Rights of Children in the Context of International Migration, para. 33, U.N. Doc. CMW/C/GC/3-CRC/C/GC/22 (16 November 2017); Comm. on the Rights of the Child, General Comment No. 6, *supra* note 5, para. 84–86. See also Pobjoy (2017).

They are frequently part of a larger set of policies and practices that seek to restrict access to asylum and place people at increased risk of serious harm. These abusive practices sometimes seek to feign compliance with international standards, for example, by claiming that those forced back were not seeking asylum or were not present on the territory. In many cases, authorities deny altogether that the practice occurs and take steps to hide their activities, including by carrying them out at night. In reality, pushbacks and other restrictions on asylum undermine the international protection framework and erode respect for the rule of law.

Solutions are not difficult to identify but are elusive to achieve. Weak oversight mechanisms, undue judicial deference to the executive, and official ambivalence are significant obstacles to the effective defense of and adherence to refugee protection. But there are hopeful signs in the form of positive domestic court rulings and other efforts to secure compliance with international standards—or at least a measure of accountability for the violations of these standards. And states are showing at least some recognition that additional avenues of protection, including for children, are needed, along with increased opportunities for safe and legal migration for those who do not meet the strict standards for asylum or other forms of protection.

## 1. Europe's New Normal

As Croatia has done, other European Union countries have also pushed back asylum seekers, sometimes using tactics that are virtually indistinguishable from those employed by Croatian border police; non-E.U. countries have done the same. To be sure, the welcome that the E.U. and other countries have extended to Ukrainians fleeing Russia's invasion is a notable exception. (See, e.g., ECRE (2023); Van Esveld (2023).) But in general, the prevalence of pushback practices is such that the Council of Europe's Commissioner for Human Rights has commented that "pushbacks and the serious violations of human rights that they entail now risk becoming a permanent and systemic feature of the way that refugees, asylum seekers and migrants are treated across Europe" (European Committee for the Prevention of Torture 2021a, p. 7).[7]

Hungary is perhaps the most brazen offender. In 2015, after Germany and other E.U. countries temporarily dropped the requirement that people seek asylum in the first E.U. country they reach,[8] Prime Minister Viktor Orbán wrote in a guest essay in the newspaper *Frankfurter Allgemeine* that Muslims threaten Europe's Christian identity. (Orbán (2015). See also Wer überrannt wird (2015) (analyzing Orbán's guest essay); Noack (2015).) Under his government, Hungary had already begun to erect barbed-wire fences along its borders with Serbia and Croatia (U.N. Special Rapporteur on Human Rights of Migrants 2020). The Orbán government introduced legal changes that authorized police to return irregular migrants to the border and direct them to exit Hungary through the fence, without any formal process.[9] Hungarian authorities have pushed back tens of thousands of people in recent years, including more than 19,000 between January and March 2022.[10] Unaccompa-

---

[7] Similarly, the Council of Europe's Parliamentary Assembly has observed that pushback "practices are widespread, and in some countries systematic" in Council of Europe member states. Council of Europe Parliamentary Assembly (2019).

[8] See, e.g., Hall and Lichfield (2015); Die Bundesregierung, Sommerpressekonferenz von Bundeskanzlerin Merkel, 31 August 2015 (announcing the change with the phrase *wir schaffen das! ("we can do this!")*, https://www.bundesregierung.de/breg-de/aktuelles/pressekonferenzen/sommerpressekonferenz-von-bundeskanzlerin-merkel-848300 (accessed on 14 February 2023). For assessments of Germany's policy of *Wilkommenkultur* ("welcoming culture") toward refugees in 2015 and 2016, see, for example, Fadaee (2021); Lemay (2021); Goldberg (2021); Oltermann (2020).

[9] See U.N. Comm. on the Rights of the Child, Concluding Observations: Hungary, paras. 38–39, U.N. Doc. CRC/C/HUN/CO/6 (3 March 2020); U.N. Comm. on the Elimination of Racial Discrimination, Concluding Observations: Hungary, paras. 22–25, U.N. Doc. CERD/C/HUN/CO/18-25 (6 June 2019; reissued 20 June 2019); U.N. Human Rights Comm., Concluding Observations: Hungary, paras. 45–50, U.N. Doc. CCPR/C/HUN/CO/6 (9 May 2018).

[10] Rendőrség (Hungarian Police), Illegális migráció alakulása—heti bontásban [Developments in illegal migration—weekly breakdown], https://www.police.hu/hu/hirek-es-informaciok/hatarinfo/illegalis-migracio-alakulasa (accessed on 14 February 2023). See also Protecting Rights at Borders Initiative (2022).

nied and separated children said that Hungarian border guards beat them, used chemical sprays on them, confiscated and destroyed their phones, forced them to remove shoes and clothing, and set dogs on them during pushbacks to Serbia between 2016 and 2018 (Save the Children 2017, 2018; Human Rights Watch 2016b).

Bulgaria has employed pushbacks since at least 2013 (Weber 2017), including of an estimated 60,000 people to Turkey in 2020 and 2021 (Tripartite Working Group 2021, 2022). In the course of pushbacks, Bulgarian police have beaten or set dogs on children and adults as standard police procedure, stolen their belongings, and stripped them of their clothes, leaving them in underwear and t-shirts before pushing them back to Turkey.[11] A 17-year-old Afghan boy told Save the Children that during one of more than 20 pushbacks he experienced, Bulgarian police ordered everyone in his group to remove their clothes and then "gathered around the fire to drink wine and made us lie naked on our backs" before forcing them to walk into Turkey (Save the Children 2022, p. 12).

"In Greece, pushbacks at land and sea borders have become de facto general policy," the UN special rapporteur on the human rights of migrants observed in April 2022.[12] Greek police, the Greek Coast Guard, and unidentified men in black or commando-like uniforms have violently pushed tens of thousands of people back to Turkey (UNHCR 2022a). In one case, a 14-year-old boy told Human Rights Watch in November 2021 that before Greek police forced him and others to cross the border into Turkey, "They took all our clothes. I was naked. No clothes.... They beat me with a police baton on my leg. I had trouble walking for the next month" (Human Rights Watch 2022b, p. 16). Human Rights Watch heard of other cases of boys ordered to strip naked, detained in overcrowded cells for hours without food or water, and then ordered to cross a river into Turkey (Human Rights Watch 2022b).

Similarly, in a series of incidents in 2020, people who had reached land after crossing the Aegean Sea were forced onto small inflatable rescue rafts without motors and cast adrift near the Turkish sea border by Greek authorities who had first stolen their personal identification, money, and other belongings. In one case of people intercepted while still at sea, the Greek Coast Guard and uniformed men in balaclavas "used dangerous maneuvers to force the boat full of migrants back to Turkey" (Human Rights Watch 2020b). In February 2022, Lighthouse Reports published accounts from "Greek officials with direct knowledge of coastguard operations" who confirmed anonymously that Greek authorities have beaten and thrown people into the sea without life jackets as a means of pushing them away from Greece (Fallon et al. 2022).

In Poland, authorities regularly take people apprehended within Polish territory back to the border and order them to cross into Belarus, ignoring requests for asylum (Human Rights Watch 2021e, 2022e; Górczyńska 2021; Amnesty International 2021). As Witold Klaus, an academic at Warsaw University's Centre for Migration Research, has observed:

> Since 2015, Polish authorities have been gradually closing borders to asylum seekers. Refugees have been approaching the Polish border applying for international protection, but the border guards "fail" to hear these requests and do not accept their applications, sending most of the asylum seekers back to Belarus. (Klaus 2021, p. 435)

Poland's "consistent practice of returning people to Belarus" has included pushbacks of children (U.N. Special Rapporteur on Human Rights of Migrants 2021). Polish border guards have also at times separated families when one family member is in need of medical

---

11  Campbell (2016); Human Rights Watch (2022c). See also Human Rights Watch (2016a) (Bulgarian police stole boots and money from and hit a 16-year-old boy before pushing him back to Turkey along with 47 others); Human Rights Watch (2014b, 17–20) (describing beatings of parents in front of their children and pushback of 16-year-old boy without shoes).

12  U.N. Special Rapporteur on Human Rights of Migrants (2022), para. 32. See also Comm. on the Rights of the Child, Concluding Observations: Greece, paras. 39(a), 40(a), U.N. Doc. CRC/C/GRC/CO/4-6 (28 June 2022); Committee against Torture, Concluding Observations: Greece, paras. 16–17, U.N. Doc. CAT/C/GRC/CO/7 (3 September 2019).

treatment; in such cases, guards have told parents to choose which will stay with a sick child while the rest of the family is pushed back to Belarus (Human Rights Watch 2021e). Latvia and Lithuania have also carried out pushbacks to Belarus, including of children travelling with their families.[13]

Spain has carried out summary expulsions, with no opportunity to make asylum claims, from Ceuta and Melilla to Morocco for many years.[14] Spanish authorities have subjected unaccompanied children as well as adults to summary expulsions,[15] known informally as *devoluciones en caliente* ("hot returns"), often employing excessive force and other ill-treatment when carrying them out.[16] Legislation enacted in 2015 allows border guards to "reject" people who attempt to enter the two cities, which are located on the coast of north Africa and share a land border with Morocco.[17] The U.N. Committee against Torture observed that the 2015 legislation "put[s] a veneer of legality on ... summary expulsion".[18]

Outside of the European Union, Turkish authorities have pushed Afghan asylum seekers to Iran and have summarily deported large numbers of Syrian asylum seekers to Syria, in some instances separating families during pushbacks (Human Rights Watch 2018a, 2021a, 2022g; C. Gall 2021).

In 2021, the E.U. Agency for Fundamental Rights noted reports of pushbacks from Austria, Cyprus, France, Latvia, Lithuania, Malta, Romania, Slovenia, and Spain, as well as from the non-E.U. countries of North Macedonia and Serbia (E.U. Fundamental Rights Agency 2021a, 2021b, 2022). These reports include summary returns from one E.U. member state to another.

In the Central Mediterranean, "pullbacks" by the E.U.-supported Libyan Coast Guard are increasingly common, and private vessels have at times carried out indirect pushbacks (Council of Europe Commissioner for Human Rights 2021; U.N. Office of the High Commissioner for Human Rights 2021). Assessing the overall situation in 2022, the Council of Europe's Commissioner for Human Rights observed that "reports of human rights violations such as denial of asylum, collective expulsions and ill-treatment associated with pushbacks pertain to at least around half of Council of Europe member states" (Council of Europe Commissioner for Human Rights 2021, p. 23).

Bilateral "readmission agreements"—arrangements under which states return people to the neighboring countries through which they have transited, with few, if any procedural safeguards[19]—can result in chain refoulement.[20] For instance, under Slovenia's readmission

---

13    Baczyńska (2021); Amnesty International (2022); Comm. against Torture, Concluding Observations: Lithuania, para. 11(f), U.N. Doc. CAT/C/LTU/CO/4 (21 December 2021).

14    See, e.g., Martínez Escamilla et al. (2014); Human Rights Watch (2014a, 2014c) (summary returns of children and adults from Melilla to Morocco).

15    See, e.g., ACCEM (2022, 30–33); Comm. on the Rights of the Child, Concluding Observations: Spain, para. 44(d), U.N. Doc. CRC/C/ESP/CO/5-6 (5 March 2018).

16    See, e.g., Human Rights Comm., Concluding Observations: Spain, para. 19, U.N. Doc. CCPR/C/ESP/CO/6 (14 August 2015).

17    Ley Orgánica 4/2000 de 11 de enero, disposición adicional décima, art. 1, BOE núm. 10 (12 January 2000) (Sp.), as amended by Ley Orgánica 4/2015, de 30 de marzo, de protección de la seguridad ciudadana, disposición final primera: Régimen especial de Ceuta y Melilla, art. 1, BOE núm. 77 (31 March 2015) (Sp.).

18    Committee against Torture, Concluding Observations: Spain, para. 13, U.N. Doc. CAT/C/ESP/CO/6 (29 May 2015).

19    See Protecting Rights at Borders Initiative (2021); Diez (2019). For an overview of the use of readmission agreements by E.U. member states, see Fitzgerald (2019).

20    "Indirect or chain *refoulement* involves removal via an intermediary country and may therefore involve the responsibility of three (or more) states." Heijer (2017, p. 485). The Committee against Torture, the Human Rights Committee, and UNHCR have recognized that the obligation not to refoule includes an obligation to ensure that a person would not face torture, threats to life or freedom, or other irreparable harm "either in the country to which removal is to be effected or in any country to which the person may subsequently be removed." Human Rights Comm., General Comment No. 31: The Nature of the General Legal Obligation Imposed on States Parties to the Covenant, para. 12, U.N. Doc. CCPR/C/21/Rev.1/Add. 13 (26 May 2004). See also Comm. against Torture, General Comment No. 4 on the Implementation of Article 3 of the Convention in the Context of Article 2, para. 12, U.N. Doc. CAT/C/GC/4 (4 September 2018) ("[T]he person at risk should never be deported to another State from which the person may subsequently face deportation to a third State in which there are substantial grounds for believing that the person would be in danger of being subjected to

agreement with Croatia, Slovenian police have summarily transferred irregular migrants to Croatia if they have entered Slovenia from Croatia, regardless of whether they have requested asylum in Slovenia (Amnesty International 2018). In turn, Croatian authorities generally push them on to Bosnia and Herzegovina or Serbia (Infokolpa 2019). Similarly, until an Italian court ordered the suspension of readmissions from Italy to Slovenia in January 2021, irregular migrants transferred to Slovenia were often immediately sent to Croatia and then pushed back to Bosnia and Herzegovina or Serbia (U.N. Special Rapporteur on Human Rights of Migrants 2021; Border Violence Monitoring Network 2020; Facchini and Rondi 2022; Gostoli 2020). The U.N. special rapporteur on the human rights of migrants has cautioned that readmission arrangements and other bilateral and multilateral agreements "cannot... be used as a strategy to bypass human rights obligations or to rubber-stamp migrant removals without individual safeguards" (U.N. Special Rapporteur on Human Rights of Migrants 2021, para. 63). In the view of UNHCR, "a reliable assessment of the risk of 'chain refoulement' must be undertaken in each individual case, prior to removal to a third country, including pursuant to a readmission agreement".[21]

Bulgaria, Croatia, Greece, and other E.U. countries are not carrying out pushbacks solely on their own initiative. These countries receive substantial E.U. funding for border management.[22] As part of the process of entry into the Schengen area—the group of 27 European countries that have removed routine mutual border controls—Croatia committed to "improvement[s] of [its] land border surveillance capacity... at the external land border with Bosnia and Herzegovina and Montenegro".[23] While the European Commission, which oversees these E.U. funds, by no means calls for or otherwise encourages pushbacks, its failure to address these abuses sufficiently (despite its border monitoring initiatives, discussed below) means that pushbacks remain largely unchecked.

Frontex, the E.U. border and coast guard agency, has been more directly implicated in pushbacks in recent years, including of children travelling alone and with their families (Fallon 2020). Frontex had detailed knowledge of Greek authorities' widespread use of pushbacks, the European Anti-Fraud Office (OLAF) concluded in a February 2022 report that did not become public until the end of July (Statius 2022a, 2022b). In the meantime, a consortium of journalists found evidence in Frontex's database of the agency's involvement in Greek pushbacks (Pascual and Statius 2022; Fallon 2022). Frontex executive director Fabrice Leggeri resigned the same day the consortium published its findings, reportedly prompted by moves by OLAF to initiate disciplinary action against him and two other Frontex officials (Pascual and Malingre 2022; Rankin 2022).

Within the European Union, the problem of state noncompliance is also compounded by competing interests within the European Commission and the European Council. For

---

torture."); Lauterpacht and Bethlehem (2003, p. 122) (prohibition of refoulement "also precludes the removal of a refugee or asylum seeker to a third State in circumstances in which there is a risk that he or she might be sent from there to a territory where he or she would be at risk").

21　UNHCR, Written Submission by the Office of the United Nations High Commissioner for Refugees in the Case of *Sharifi and Others v. Italy and Greece* (Application No. 16643/09), at 2 (October 2009), https://www.refworld.org/pdfid/4afd25c32.pdf (accessed on 14 February 2023).

22　Eur. Comm'n, Managing Migration: E.U. Financial Support to Greece, January 2021 (€3.12 billion in E.U. funding for migration management between 2015 and end of 2020), https://home-affairs.ec.europa.eu/system/files/2021-01/202101_managing-migration-eu-financial-support-to-greece_en.pdf (accessed on 14 February 2023); Eur. Comm'n, Managing Migration: E.U. Financial Support to Bulgaria, Feb. 2020 (€320.4 million in E.U. funding for migration management between 2015 and early 2020), https://home-affairs.ec.europa.eu/system/files/2020-02/202002_managing-migration-eu-financial-support-to-bulgaria_en.pdf (accessed on 14 February 2023); Eur. Comm'n, Managing Migration: E.U. Financial Support to Croatia (January 2021) (€163.13 million in E.U. funding for migration management between 2015 and end of 2020), https://home-affairs.ec.europa.eu/system/files/2021-01/202101_managing-migration-eu-financial-support-to-croatia_en.pdf (accessed on 14 February 2023).

23　Eur. Comm'n, Commc'n from the Comm'n to the Eur. Parliament and the Council on the Verification of the Full Application of the Schengen *Acquis* by Croatia, COM(2019) 497 final, at 6 (22 October 2019), https://www.europarl.europa.eu/RegData/docs_autres_institutions/commission_europeenne/com/2019/0497/COM_COM(2019)0497_EN.pdf (accessed on 14 February 2023). See also Eur. Council, Council Conclusions on the Fulfilment of the Necessary Conditions for the Full Application of the Schengen *Acquis* in Croatia, Doc. 14883/21 (9 December 2021).

instance, the expansion of the (generally) border-free Schengen area furthers the free movement of people and goods, two of the fundamental principles of the European Union.[24] But it is difficult to reconcile the European Commission's conclusion in 2019 that Croatia was ready to join the Schengen area[25]—a process that concluded with Croatia's entry into the area in January 2023[26]—with Croatia's track record of serious human rights violations against migrants, particularly in light of the Schengen Borders Code requirement that E.U. member states "act in full compliance with ... relevant international law," including the Refugee Convention, "obligations related to access to international protection, in particular the principle of non-refoulement, and fundamental rights".[27]

As Iris Goldner Lang and Boldizsár Nagy have observed, "the gap between the E.U.'s constitutional, normative expectations and member states' practices has been widening, leading to tension between the reality and the law" (Lang and Nagy 2021). And as the example of Croatia's admission to the Schengen zone illustrates, E.U. institutions have contributed to these tensions between law and practice.

## 2. A Worldwide Phenomenon

Pushbacks are all too common elsewhere in the world. The U.N. special rapporteur on the human rights of migrants has received reports of large-scale pushbacks from Algeria to Niger, in some cases separating children from their families in the course of expulsions, as well as pushbacks from Libya to Chad, Egypt, and Sudan; from Mexico to Guatemala and from Guatemala to Honduras; from Trinidad and Tobago to Venezuela; and from Iran to Afghanistan, among other contexts (U.N. Special Rapporteur on Human Rights of Migrants 2021). The Committee on the Rights of the Child has noted "[i]ncreasing pushbacks of Venezuelan children and systematic rejections of asylum petitions at the border" by Chile.[28]

This is by no means an exhaustive list. For example, in April 2020, Houthi forces expelled thousands of Ethiopian migrants from northern Yemen to the Saudi border, where Saudi border guards then opened fire, killing dozens (Human Rights Watch 2020c). Pushbacks from Malaysia left hundreds of Rohingya in life-threatening conditions at sea in 2020 (Human Rights Watch 2020a; Médecins Sans Frontières 2020). Indonesia and Thailand have also pushed back boats with Rohingya and other asylum seekers and migrants (Human Rights Watch 2015). Children, alone and with their families, are often subject to pushbacks in these and other situations (U.N. Special Rapporteur on Human Rights of Migrants 2021).

Australia developed what are perhaps the most systematic pushback policies. It began to intercept boats in late 2001 under a policy known as "Operation Relex" (Senate Select Committee 2002, 13–30), turning back five boats carrying a total of 614 people between October 2001 and November 2003. (Spinks (2018, 2). See also Kaldor Centre (2018); Schloenhardt and Craig (2015).) Boat "turnbacks"—a term commonly used in Australia to describe the practice, though it is interchangeable with "pushbacks"—resumed in September 2013, when a policy known as "Operation Sovereign Borders" included an explicit commitment to turn back boats arriving irregularly in Australian waters "where it

---

[24]　See Treaty on the Functioning of the European Union arts. 28, 45, in Consolidated Version of the Treaty on the Functioning of the European Union, O.J. L. 326/47-326/390 (26 October 2012), https://eur-lex.europa.eu/legal-content/EN/TXT/?uri=celex%3A12016ME%2FTXT (accessed on 14 February 2023).

[25]　Eur. Comm'n, Commc'n from the Comm'n to the Eur. Parliament and the Council, *supra* note 23, at 14 ("The Commission considers that Croatia has taken the measures needed to ensure that the necessary conditions for the application of all relevant parts of the Schengen acquis are met.").

[26]　See Kusmanovic and Timu (2022). For background on the final stages of the process, see *Schengen Area: Council Requests European Parliament's Opinion on a Draft Decision on the Full Application of the Schengen Acquis in Croatia*, Eur. Council (29 June 2022), https://www.consilium.europa.eu/en/press/press-releases/2022/06/29/espace-schengen-le-conseil-demande-l-avis-du-parlement-europeen-sur-un-projet-de-decision-relative-a-la-pleine-application-de-l-acquis-de-schengen-en-croatie/ (accessed on 14 February 2023).

[27]　Regulation (EU) 2016/399 of the Eur. Parliament and of the Council of 9 March 2016 on a Union Code on the Rules Governing the Movement of Persons Across Borders (Schengen Borders Code) art. 4, O.J. L 77/1 (26 March 2016).

[28]　Comm. on the Rights of the Child, Concluding Observations: Chile, para. 34(e), U.N. Doc. CRC/C/CHL/CO/6-7 (22 June 2022).

is safe to do so".[29] Australia turned back at least 33 boats carrying more than 800 people, including at least 130 children, between September 2013 and May 2022 (Refugee Council of Australia 2022; Borys 2022). Australia's new government has said it will continue Operation Sovereign Borders and boat turnbacks, and indeed carried out turnbacks immediately after taking office (Galloway 2022; Hevesi 2022).

The United States created and implemented what was likely the first modern push-back policy. Under the Haitian interdiction policies first implemented by the Reagan administration in 1981 (Legomsky 2006; Frelick 1988, 1993, 2004), the U.S. Coast Guard boarded Haitian boats in international waters "to enforce the suspension of the entry of undocumented aliens".[30] By 1990, the United States had interdicted 364 boats, allowing just six of the people on board to apply for asylum and returning more than 21,000 others to Haiti (Lawyers Committee for Human Rights 1990), often after holding them for a time at the U.S. naval base at Guantánamo Bay, Cuba. A 1992 executive order by President George H. W. Bush eliminated nonrefoulement screening altogether,[31] an approach upheld by the U.S. Supreme Court the following year.[32] President Bill Clinton continued the no-screening policy until 1994, when U.S. officials resumed screenings (Frelick 2004). As in earlier years, screenings rarely resulted in protection for interdicted Haitians: in 1994 and 1995, for example, U.S. authorities forcibly returned nearly all of the more than 20,000 Haitians held at Guantánamo Bay, including more than 300 unaccompanied children (Little 1999). Bush and Clinton administration policies also provided for "offshore processing" and "safe haven" in third countries, although U.S. authorities made few actual transfers to third countries (Dastyari 2013).

Interdictions at sea were not limited to Haitians. After Cuba lifted exit restrictions in 1994 in response to escalating protests, President Clinton announced that the Coast Guard would "continue its expanded effort to stop any boat illegally attempting to bring Cubans to the United States".[33] The Coast Guard intercepted more than 37,000 Cubans in 1994, more than ten times the number of Cubans interdicted in 1993 (Fullerton 2004), transferring some 8500 to a U.S. air force base in Panama and about 28,000 to Guantánamo Bay (Sartori 2001). Dominicans and other nationalities have also been subject to Coast Guard interdictions (Legomsky 2006).

The United States continues to maintain its policy of maritime interdictions, reinforced by a 2002 executive order authorizing the Secretary of Homeland Security to detain and screen undocumented people intercepted in the Caribbean "in Guantanamo Bay Naval Base or any other appropriate location".[34] The Coast Guard reports that it interdicted more than 7100 Haitians, 6100 Cubans, and 1700 Dominicans between October 2021 and September 2022.[35] From October 2022 through January 2023, the first five months of fiscal year 2023, the Coast Guard interdicted more than 5300 Cubans (Groll 2023).

More recently, the United States has also carried out what may be the largest number of pushbacks, at least on an annual basis. Since March 2020, the U.S. government has used the pretext of the COVID-19 pandemic to block access to asylum and instead summarily expel people apprehended at or near the land border with Mexico.[36] Under the policy,

---

29 See Kaldor Centre (2018). For a discussion of whether Australia's turnback policies blurred or crossed legal lines, see Missbach and Palmer (2020).

30 Exec. Order No. 12,324, § 2(a), 46 Fed. Reg. 48,109, 48,109 (29 September 1981).

31 Exec. Order No. 12,807, 57 Fed. Reg. 23,133 (1 June 1992).

32 Sale v. Haitian Centers Council, 509 U.S. 155 (1993) (concluding that the nonrefoulement obligations in the Refugee Convention and U.S. law did not apply on the high seas). For an assessment of the Supreme Court's decision, see Koh (1994). For an insightful discussion of the legacies and counter-legacies of *Sale,* see Koh (2014).

33 See Excerpts from News Conference (1994) ("I have directed the Coast Guard to continue its expanded effort to stop any boat illegally attempting to bring Cubans to the United States").

34 Exec. Order No. 13,276, 67 Fed. Reg. 69,985 (19 November 2002).

35 For more details, see Shepard (2022); Boyette (2023); Coast Guard Repatriates 47 Dominicans (2022).

36 See Beckett et al. (2022); Cayanan (2022). For an assessment of the legal and practical challenges the Biden administration has faced in its efforts to modify the policy, see Biden Administration Continues Efforts (2022).

often referred to as "Title 42,"[37] U.S. Customs and Border Protection (CBP) agents need not allow people to make asylum claims, do not generally consider stated fears of or likely danger of harm if sent to Mexico or home countries, and do not permit people to contact an attorney or see a judge (Pillai and Artiga 2022). Instead, border agents have taken most people to the nearest land border crossing and ordered them to walk into Mexico, even though many of those expelled are not Mexican nationals (Human Rights Watch 2021b). In other cases, U.S. officials have sent people directly to their countries of origin—including Haiti, despite the humanitarian, political, and human rights crisis in the country (Human Rights Watch 2022d, 2022f). Human Rights First (2022) tracked nearly 10,000 violent attacks on people between March 2020 and March 2022, including cases of kidnapping, torture, and rape, after they were summarily expelled. These summary expulsions have had a disparate impact on migrants of African and Latino descent, as the U.N. Committee on the Elimination of Racial Discrimination observed in August 2022.[38]

Unaccompanied children were subject to summary expulsion under the policy until February 2021; families with children continued to be subject to summary expulsion as of January 2023. When the Mexican state of Tamaulipas, which includes the border towns of Matamoros and Laredo, told CBP it would not accept expulsions of families with children under the age of seven, CBP began carrying out "'lateral' transfers by plane or bus to other locations along the border, such as El Paso, where Mexican authorities will allow the agency to expel families with young children," according to the American Immigration Council (2022).

At least 8800 unaccompanied children were summarily expelled in the first six months the policy was in effect (Montoya-Galvez 2020). By the end of December 2022, U.S. border agents had carried out nearly 16,000 summary expulsions of unaccompanied children.[39] Counting adults as well as children, U.S. authorities had conducted more than 2.5 million summary expulsions through the end of December 2022.[40]

In addition, under a pre-pandemic policy known as "metering," CBP agents turned back thousands of people, including families with children, at the U.S.–Mexico border, responding to requests for asylum with variations on the responses "there is no processing capacity" (Leutert et al. 2018, p. 4), the border post is "too full," (Human Rights Watch 2019) or "Trump says we don't have to let you in" (Human Rights First 2017, p. 5). Some unaccompanied children were also turned away from border stations under the "metering" policy (Human Rights Watch 2018b).

## 3. Increased Danger and Insecurity

Pushbacks do not dissuade people from seeking safety, but they raise the cost and risk of doing so.

There is no evidence, for example, that the United States' summary expulsion policy has reduced the number of people who try to enter irregularly. To the contrary, "it has produced record levels of 'recidivism'—repeat, irregular crossings—among migrants who

---

[37] The public health law used as the basis for the summary expulsions order is codified at title 42 of the United States Code. Summary expulsions were authorized by the U.S. Centers for Disease Control and Prevention, not the immigration enforcement agencies in the U.S. Department of Homeland Security. See 42 U.S.C. §§ 265, 268 (codifying sections 362 and 365 of the Public Health Service Act of 1944, Pub. L. No. 78-410); Control of Communicable Diseases; Foreign Quarantine: Suspension of Introduction of Persons into United States from Designated Foreign Countries or Places for Public Health Purposes, 85 Fed. Reg. 16,559 (24 March 2020).

[38] Comm. on the Elimination of Racial Discrimination, Concluding Observations: United States of America, para. 51(c), U.N. Doc. CERD/C/USA/CO/10-12 (30 August 2022).

[39] U.S. Customs & Border Protection, Southwest Land Border Encounters, https://www.cbp.gov/newsroom/stats/southwest-land-border-encounters (accessed on 14 February 2023) (select "all" for fiscal years, "all" for components, "UC/single minors" for demographics, "Title 42" for authority). Although it is also possible to view results for children travelling with their families, the results are not complete; they show only summary expulsions carried out by the Office of Field Operations (the agency that staffs official border crossings), not those by the U.S. Border Patrol. In September 2020, the Trump administration estimated that border agents expelled "approximately 7600 members of migrant families with children." Montoya-Galvez (2020).

[40] U.S. Customs & Border Protection, Southwest Land Border Encounters, *supra* note 39 (select "all" for fiscal years, "all" for components, "all" for demographics, "Title 42" for authority).

have no option but to attempt entry without inspection," write My Khanh Ngo and Shaw Drake, lawyers with the American Civil Liberties Union (Ngo and Drake 2022).

Evidence from other contexts of migration suggests that restrictive or harsh government policies may not be significant factors in people's decisions to leave their countries of origin or their choice of destination.[41] Instead, considerations such as perceptions of safety, freedom, and opportunity; family ties and other support; expectations of and obligations toward parents; and in some contexts cultural and linguistic ties in the intended destination are frequently far more influential. (E.g., Belloni (2019, pp. 3, 79, 91–95, 140–41). See also Yıldız (2021).) People may also not be aware of government policies or may choose to believe that the policies will not affect them.

As with border walls (see, e.g., R. Jones (2016); Slack et al. (2016)), pushbacks and related policies may, however, affect peoples' route and means of travel (see, e.g., Kuschminder et al. (2015, pp. 51–52); Bobić and Santić (2020)), often increasing the risks they face. For instance, research suggests that increased reliance on smugglers, together with increased vulnerability to exploitation and other harm, was a consequence of the E.U.–Turkey deal. (See, e.g., Yıldız (2021, pp. 147–48, 151–52).) The same is true in other contexts, as the U.N. special rapporteur on torture has observed: "In practice, border closures" by means of fences, walls, or the like "tend to encourage smuggling, crime and police corruption, and to expose irregular migrants to extortion, violence, sexual abuse and the risk of trafficking" (U.N. Special Rapporteur on Torture 2018, para. 50).

Pushbacks are better understood as exercises in symbolism, part of what some researchers have termed the "politics of spectacle" (Cantat 2020). As in Australia, Hungary, the United States, and other countries, the "border spectacle" (De Genova 2013, 2015) involves the staging of "dramatic scenes of enforcement at/of the border" for the purpose of "displaying the power of the state to enforce the politics of exclusion and control on which national authority and sovereignty rely" (Cantat 2020, pp. 184–85).

Even where, as in Bulgaria, Croatia, and Greece, authorities take efforts to keep their abuses clandestine and disclaim responsibility when their practices become public, there is an element of performance in what is often a carefully controlled narrative intended to signal that the government is taking tough measures against irregular migration.[42]

## 4. One Component of an Abusive Agenda

Pushbacks are often one element in a larger strategy that targets irregular migration without regard for protection needs. Selectively or in combination, states have employed additional measures that restrict access to asylum and other protection, including for children.

### 4.1. Emergency Restrictions

The use of emergency measures is one tactic. This approach is not necessarily abusive in itself—human rights treaties recognize that emergencies can require considered, proportionate responses.[43] But "emergency" (usually security, more recently health) is all too often invoked as cover for overreach.

The public health order authorizing U.S. summary expulsions, discussed above, is one example: as a group of epidemiologists and public health experts wrote in January 2022,

---

[41] See, e.g., Hiskey et al. (2018); Ryo (2019); Cox and Goodman (2018); Czaika and Hobolth (2016). But see Matsui and Raymer (2020). Similarly, there is little evidence that media campaigns aimed at deterring irregular migration are effective. See, e.g., Musarò and Migrants (2019); Brown (2015).

[42] See, e.g., Kreizer (2019) (quoting Interior Minister Davor Bozinovic's June 2018 statement that "Croatia... has the strongest border police in this part of Europe"); Helms (2022).

[43] See, e.g., The Siracusa Principles on the Limitation and Derogation of Provisions in the International Covenant on Civil and Political Rights, in Permanent Rep. of the Neth. to the U.N., Note verbale dated 24 August 1984 from the Permanent Rep. of the Neth. to the U.N. Office at Geneva addressed to the Secretary-General, annex, U.N. Doc. No. E/CN.4/1985/4 (28 September 1984); Svensson-McCarthy (1998), 147, 195–99. See also ECRE (2020a).

the "order is nothing more than a politically expedient measure that exploits the COVID-19 pandemic to expel or block from the country people seeking asylum".[44]

As another example, Hungary declared a "state of crisis due to mass migration" in September 2015,[45] which it extended twelve times, through September 2022 (Hungarian Helsinki Committee 2022a). Hungarian authorities used the "state of crisis" to require most asylum applications to be submitted in the border "transit zones" and required applicants, including unaccompanied children of age 14 and above, to remain in the transit zones throughout the asylum procedure. (Hungarian Helsinki Committee (2022a), pp. 20–21. See also U.N. Special Rapporteur on Human Rights of Migrants (2020), paras. 25–27, 30, 34.) After the Court of Justice of the European Union ruled in May 2020 that the transit zones were a form of unlawful detention,[46] Hungary closed the zones and eliminated access to asylum from within the country and at the border (ECRE 2020b, 2020c). Hungary now requires most people to initiate asylum applications at either the Belgrade or Kyiv embassy (Hungarian Helsinki Committee 2022a). By the end of December 2021, only 12 people had managed to file asylum applications under the new procedures (Hungarian Helsinki Committee 2022b).

Lithuania and Slovenia, among other countries, have also passed new legislation allowing authorities to restrict access to asylum in the event of a "mass influx of foreigners," a "complex crisis in the field of migration," or similar conditions (Amnesty International 2022; PIC 2022).

Moreover, many of the 27 European countries in the Schengen area have periodically invoked emergencies to temporarily reintroduce "internal" border controls, including in response to the movement of refugees in 2015 and 2016 (Guild 2021; Thym and Bornemann 2020; Ceccorulli 2019). Even when other grounds are the basis for reintroduced controls, they can be misused to restrict access to asylum and other protection. In France, for example, where authorities have regularly renewed border controls as a counterterrorism measure since November 2015, border police have often circumvented child protection safeguards and have detained unaccompanied children with unrelated adults (Human Rights Watch 2021c; Contrôleur général 2018).

Worldwide, some 100 countries initially restricted access to asylum in response to the COVID-19 pandemic. In May 2022, more than 20 countries continued to deny or restrict access to asylum on public health grounds while in nearly all cases reopening their borders to tourism and business travel. (UNHCR (2022b). See also Ghezelbash and Tan (2020).)

### 4.2. "Safe Third Country" Designations

The use of "safe third country" (or even "safe country of origin") designations may foreclose asylum claims without individualized consideration, and often without a meaningful assessment of the situation in the purportedly "safe" country. These designations may, therefore, camouflage collective expulsions and refoulement.

For instance, under the pretext that Serbia was a "safe third country," Hungary rejected nearly all asylum applications, often on the day of their submission, between 2015 and 2020 (Hungarian Helsinki Committee 2022b). Under the 2016 E.U.–Turkey deal, the European Union regards Turkey as a "safe third country" for asylum seekers even though Turkey ratified the Refugee Convention with the proviso that non-Europeans can only receive

---

[44] Letter from Megan Coffee, Infectious Disease Specialist and Assistant Professor, Columbia Univ. Mailman Sch. of Pub. Health, et al., to Xavier Becerra, Sec'y, U.S. Dep't of Health and Human Servs., et al. (28 January 2022), https://www.publichealth.columbia.edu/research/program-forced-migration-and-health/january-2022-letter-vaccination-southern-border (accessed on 14 February 2023). See also Gilman (2020).

[45] 2015. évi CXL. törvény a Egyes törvényeknek a tömeges bevándorlás kezelésével összefüggő módosításáról szóló (Law No CXL of 2015 amending certain laws in the context of managing mass immigration), Magyar Közlöny (Hungarian Gazette) 2015/124 (7 September 2015) (Hung.), https://magyarkozlony.hu/dokumentumok/e3c72d64adc98b5c5b961da1c09a9069158b80d7/megtekintes (accessed on 14 February 2023).

[46] Joined Cases C-924/19 PPU and C-925/19 PPU, FMS et al. v. Országos Idegenrendészeti Főigazgatóság Dél-alföldi Regionális Igazgatóság and Országos Idegenrendészeti Főigazgatósá [GC], ECLI:EU:C:2020:367 (CJEU 14 May 2020).

"conditional" or temporary protection (Terry 2021; Lehner 2019). An "Asylum Cooperative Agreement" in effect between November 2019 and January 2021 led to the transfer of more than 900 people from the United States to Guatemala, despite ample indications that Guatemala lacked the capacity to adjudicate asylum claims at that scale (Senate Foreign Relations Committee Democratic Staff 2021). And an agreement between the United Kingdom and Rwanda may result in people being sent to Rwanda with the possibility of pursuing asylum there, but not in the United Kingdom,[47] notwithstanding warnings by UNHCR and human rights groups that people transferred to Rwanda would be at risk of serious harm and that Rwanda lacked the capacity to process asylum claims.[48]

A similar logic underlay the U.S. "Remain in Mexico" program, which sent some 71,000 people, including more than 20,000 children, to border towns in Mexico while their U.S. asylum claims were processed (Human Rights Watch 2022a). Among other foreseeable harms, more than 340 children were kidnapped or faced a kidnapping attempt after they were placed in the program with their families (Human Rights First 2021).

*4.3. Other Measures*

"Externalized" migration controls can include, in addition to pushbacks and other approaches discussed above, financial or other incentives for countries of origin and third countries to impede migration, carrier sanctions, and other requirements imposed on the private sector, as well as other measures, as Bill Frelick, Ian Kysel, and Jennifer Podkul have observed (Frelick et al. 2016).

Accelerated asylum procedures (see, e.g., ECRE (2017); McDonald and O'Sullivan (2018); Peralta (2015)), restrictive interpretations of standards for refugee recognition,[49] and destitution or similar measures[50] are also part of the playbook that states often follow.

These are not new tactics: a 1999 report by the Council of Europe's Committee on Migration, Refugees and Demography identified many of these and other trends and warned that they threatened "the generous vision and human rights values, including freedom from persecution, that inspired [the council's] creation".[51] But as the U.N. special rapporteur on torture observed in 2018, "an escalating cycle of repression and deterrence"

---

[47]    Home Office, Memorandum of Understanding Between the Government of the United Kingdom of Great Britain and Northern Ireland and the Government of the Republic of Rwanda for the Provision of an Asylum Partnership Arrangement (14 April 2012), https://www.gov.uk/government/publications/memorandum-of-understanding-mou-between-the-uk-and-rwanda/memorandum-of-understanding-between-the-government-of-the-united-kingdom-of-great-britain-and-northern-ireland-and-the-government-of-the-republic-of-r (accessed on 14 February 2023). See also Home Office, UK and Rwanda Migration and Economic Development Partnership (14 April 2022) ("We have agreed that people who enter the UK illegally will be considered for relocation to Rwanda to have their asylum claim decided. And those who are resettled will be given support, including up to five years of training to help with integration, accommodation, and healthcare, so that they can resettle and thrive."), https://www.gov.uk/government/speeches/home-secretarys-speech-on-uk-and-rwanda-migration-and-economic-development-partnership (accessed on 14 February 2023); Terry (2022).

[48]    See, e.g., Casciani (2022). In June 2022, the European Court of Human Rights ordered the United Kingdom not to remove an asylum seeker to Rwanda until his legal challenge was concluded in the domestic courts. European Court of Human Rights, *The European Court Grants Urgent Interim Measure in Case Concerning Asylum-Seeker's Imminent Removal from the UK to Rwanda*, 14 June 2022.

[49]    See, e.g., Pannia (2021); Ahmetašević (2020); Greek Council for Refugees and Oxfam (2020) (noting that the 2020 Greek asylum law has limited access to asylum).

[50]    For example, in 2017 Hungary stopped giving food to asylum seekers in a camp primarily housing people whose cases were on appeal. Cantat (2020, p. 190). In the United Kingdom, asylum-seeking families have been housed in inadequate and sometimes dangerous temporary accommodations and faced barriers to access to essential health services (L. Jones et al. 2022), and undocumented families, many with children who have spent all or most of their lives in the country, have faced extreme poverty as the result of "hostile environment" policies (Coram Children's Legal Centre 2013; Schweitzer 2020). More generally, restrictions on the right of asylum seekers to work often place considerable pressure on households. See, e.g., Burchett and Matheson (2010); Valenta and Thorshaug (2013). See also Fasani et al. (2020) (finding that employment restrictions have "potentially large costs for both affected refugees and hosting societies"); Marbach et al. (2018) (finding considerable cost to taxpayers of such restrictions on work).

[51]    Council of Europe, Comm. on Migration, Refugees and Demography, Restrictions on Asylum in the Member States of the Council of Europe and the European Union, summary, Doc. 8598 (21 December 1999).

([U.N. Special Rapporteur on Torture 2018](#), para. 7) now characterizes many states' approach to irregular migration, including by those who seek safety.

Such approaches are also at odds with states' political commitment in the Global Compact on Migration to expanded pathways for safe and legal migration channels[52] and in the parallel Global Compact on Refugees to strengthen international cooperation and solidarity to protect refugees.[53]

## 5. Positive Developments, with Some Setbacks

The prevalence of pushbacks and related abuses is discouraging, particularly in light of their condemnation by the Council of Europe's Parliamentary Assembly and Commissioner for Human Rights, the U.N. special rapporteurs on torture and on the human rights of migrants, and other authorities.

Pushbacks may result in the return of people, directly or indirectly, to places where their lives or freedom would be threatened or where they would face torture, in violation of the principle of nonrefoulement.[54]

Expulsions of children to places where they face "a real risk of irreparable harm"[55] or which are not based on "a robust individual assessment and determination of the best interests of the child"[56] violate states' obligations under the Convention on the Rights of the Child. Accordingly, the Committee on the Rights of the Child has criticized and called for an end to pushbacks by authorities in Greece, Croatia, Cyprus, Spain, and other countries.[57]

In addition, under the jurisprudence of the European Court of Human Rights, the prohibition of collective expulsions[58] is violated when states remove people "without examining their personal circumstances and, consequently, without enabling them to put forward their arguments against the measure taken by the relevant authority".[59] Pushbacks, which prevent people from reaching, entering, or remaining in a particular territory and usually afford summary or no screening for protection needs, readily meet this standard.

Moreover, as discussed above, pushbacks are frequently carried out with excessive force, may separate families, and subject people to other forms of ill-treatment, among other violations.

---

[52] Global Compact for Safe, Orderly and Regular Migration, objective 5, G.A. Res. 73/195, U.N. Doc. A/RES/73/195 (11 January 2019).

[53] Global Compact on Refugees, in Rep. of the U.N. High Comm'r for Refugees, pt. II, GAOR, 73d sess., Supp. No. 12, U.N. Doc. A/73/12 (part II) (2 August 2018). For an assessment of the potential of the two compacts, see McAdam (2018). But see Hathaway (2018); Aleinikoff (2018); Chimni (2018).

[54] The principle of nonrefoulement is reflected in treaty obligations and is also a norm of customary international law. Convention relating to the Status of Refugees art. 33(1), 28 July 1951, 189 U.N.T.S. 137; Convention against Torture, *supra* note 5, art. 3; European Convention on Human Rights, *supra* note 5, art.3; Charter of Fundamental Rights of the European Union *supra* note 4, arts. 4, 19(2); UNHCR, Advisory Opinion on the Extraterritorial Application of Non-Refoulement Obligations Under the 1951 Convention relating to the Status of Refugees and Its 1967 Protocol, paras. 14–16 (26 January 2007).

[55] Comm. on the Rights of the Child, General Comment No. 6, *supra* note 5, para. 27.

[56] Joint General Comment No. 3 (Comm. on Migrant Workers) and No. 22 (Comm. on the Rights of the Child), *supra* note 6, para. 33.

[57] See, e.g., Comm. on the Rights of the Child, Concluding Observations: Greece, *supra* note 12, paras. 39(a), 40(a); c. Comm. on the Rights of the Child, Concluding Observations: Cyprus, paras. 37(a), 38(a), U.N. Doc. CRC/C/CYP/CO/5-6 (24 June 2002), Comm. on the Rights of the Child, Concluding Observations: Croatia, para. 40 U.N. Doc. CRC/C/HRV/CO/5-6 (22 June 2022); Comm, on the Rights of the Child, Concluding Observations: Chile, para. 35(h), U.N. Doc. CRC/C/CHL/CO/6-7 (22 June 2022); Comm. on the Rights of the Child, Concluding Observations: Spain, *supra* note 15, para. 45(d). See also Comm. on the Rights of the Child, Concluding Observations: Hungary, *supra* note 9, para. 39(a) (calling for prohibition on the immediate expulsion of children and their families who are in irregular status and have not had the opportunity to apply for asylum); D.D. v. Spain, Commc'n No. 4/2016, paras. 14.5–14.9, Comm. on the Rights of the Child, U.N. Doc. CRC/C/80/D/4/2016 (15 May 2019) (finding that the immediate return of an unaccompanied child from Spain to Morocco without assessing risk of irreparable harm, taking into account best interests, giving the child an opportunity to challenge deportation, or affording special protection and assistance violated articles 3, 20, and 37 of the Convention on the Rights of the Child).

[58] Protocol No. 4, *supra* note 4, art. 4 ("Collective expulsion of aliens is prohibited.").

[59] Hirsi Jamaa & Others v. Italy [GC], para. 177, 2012-II Eur. Ct. H.R. 97. See also Sharifi et autres c. Italie et Grèce [GC], paras. 214–25, Requête No. 16643/09 (Eur. Ct. H.R. 21 October 2014) (automatic returns to Greece deprived migrants of any effective possibility of receiving asylum).

In a sign that E.U. institutions recognize that pushbacks at the external borders are inconsistent with regional and international obligations, a 2020 proposal by the European Commission would require E.U. member states to establish independent border monitoring mechanisms.[60] Commenting on the proposal, a group of eight nongovernmental organizations have called for an expanded scope of the mechanisms, guarantees for their independence, strengthened accountability for violations, and explicit consequences for member states that obstruct or disregard the mechanisms.[61] The European Committee for the Prevention of Torture (2021b) has called for any new monitoring mechanisms to meet specific criteria of effectiveness and independence, including unfettered access to border areas without notice, to relevant documentation, and to alleged victims of violations and the authority to engage directly with prosecutors' offices, as well as others with information relevant to its investigations.

In the absence of essential guarantees of effectiveness and independence, Croatia's border monitoring mechanism, established in 2021 with E.U. funding, has so far failed to live up to its promise (Human Rights Watch 2021d). The mechanism's mandate appears to be limited to police stations around the border, border crossing points, and detention centers. It lacks the authority to conduct unannounced inspections and does not have access to the Croatian Ministry of the Interior's databases (Centre for Peace Studies 2022). Despite these significant limitations, the initial version of its first report, published in December 2021, noted that "police carry out unlawful deterrence (pushbacks)," among other "irregularities in police conduct".[62] This version disappeared from the government website the following day and was replaced a week later with a new version describing pushbacks as "isolated cases".[63]

Assessing Croatia's border monitoring mechanism and the European Commission's oversight of the E.U. funding used to establish it, the E.U. ombudsman found "significant shortcomings … as regards how fundamental rights compliance was monitored" and called on the European Commission to "take an active role in overseeing the monitoring mechanism and demand concrete and verifiable information from the Croatian authorities on the steps taken to investigate reports of collective expulsions and mistreatment of migrants and asylum seekers".[64]

In light of these shortcomings, it is not surprising that Croatia's monitoring mechanism "has not ended pushbacks," a Protecting Rights at Borders Initiative report concluded in January 2023. In fact, the report cautioned that the mechanism's ineffectiveness "might wrongfully send the signal that Croatia has been improving their procedures, while rights violations at borders continue as a daily practice" (Protecting Rights at Borders Initiative 2023, p. 1).

---

[60] Eur. Comm'n, Proposal for a Regulation of the European Parliament and of the Council Introducing a Screening of Third Country Nationals at the External Borders and Amending Regulations (EC) No. 767/2008, (EU) 2017/2226, (EU) 2018/1240 and (EU) 2019/817, art. 7 & pmbl. para. 23, Doc. COM(2020) 612 final (23 September 2020).

[61] European Council on Refugees and Exiles (ECRE) (2020d). See also Lanneau (2021) (concluding that "[t]he proposal of the Commission to create a new monitoring independent mechanism falls short of expectations").

[62] Independent Monitoring Mechanism, First Half-Year Report of the Independent Mechanism for Monitoring the Conduct of Police Officers of the Ministry of the Interior in the Field of Irregular Migration and International Protection, June–December 2021 (December 2021) [working version published 3 December 2021], https://www.cms.hr/system/article_document/doc/763/Working_version_of_the_1st_IBMM_report.pdf (accessed on 14 February 2023).

[63] Independent Monitoring Mechanism, First Semi-Annual Report of the Independent Oversight Mechanism Monitoring the Actions of Police Officers of the Ministry of the Interior in the Field of Irregular Migration and International Protection, June-December 2021 (December 2021) [final version published 10 December 2021], https://www.cms.hr/system/article_document/doc/764/Final_version_of_the_1st_IBMM_report.pdf (accessed on 14 February 2023). See also Centre for Peace Studies (2021).

[64] Decision Concerning How the European Commission Monitors and Ensures Respect for Fundamental Rights by the Croatian Authorities in the Context of Border Management Operations Supported by EU Funds, Case No. 1598/2020/VS, Eur. Ombudsman (22 February 2022), https://www.ombudsman.europa.eu/en/decision/en/152811 (accessed on 14 February 2023).

Particularly in the absence of robust monitoring mechanisms, regional litigation is an obvious approach for a measure of accountability for the human rights violations caused by and during pushbacks.

But recent European Court of Human Rights caselaw risks undermining what has been a robust legal standard on collective expulsions. In a 2020 decision, *N.D. and N.T. v. Spain*, the court's Grand Chamber developed an exception to the prohibition of collective expulsions for "situations in which the conduct of persons who cross a land border in an nauthorized manner, deliberately take advantage of their large numbers and use force, is such as to create a clearly disruptive situation which is difficult to control and endangers public safety".[65] In a subsequent decision, the court has appeared to broaden the exception to apply presumptively to other situations in which "by crossing the border irregularly, the applicants circumvented an effective procedure for legal entry".[66]

This "own culpable conduct" exception has been rightly criticized.[67] As Vera Wriedt observes, "[t]he Court considered that these conditions applied in the case of *N.D. and N.T.* (although violence was perpetrated by the authorities rather than by the claimants) and blamed the applicants for not using legal pathways (despite the substantive evidence showing that these were not available in practice), portraying them as a danger (rather than in danger)" (Wriedt 2022). Three of the court's judges have called for the rule announced in *N.D. and N.T.* to be "confined to its proper context in order to avoid depriving the right [to protection from collective expulsion] of its very essence".[68]

Even so, the exception did not prevent the court from recent findings that pushback practices by Croatia, Hungary, and Poland violated the prohibition of collective expulsion as well as the right to protection from ill-treatment and the right to an effective remedy.[69]

Pushback cases before the court can, of course, hold states to account without raising claims of collective expulsion. In a 2022 judgment against Greece, the European Court of Human Rights examined a likely pushback operation during which a fishing boat that was transporting 27 people from Turkey to Greece capsized. Eleven people, some of them children, died. Survivors said their boat sank due to actions by a Greek coast guard speedboat trying to drive them back to Turkey; the Greek government blamed the boat's passengers for panicking during a rescue attempt, though it was not clear why a speedboat with no life jackets or other rescue equipment would be attempting a rescue without support. The court found that Greek security officials' actions during a likely pushback operation failed to safeguard life, subjected many of the people on the boat to degrading treatment after their rescue, and then failed to carry out a thorough investigation.[70] At least 32 other cases involving pushbacks by Greek authorities were before the European Court of Human Rights as of July 2022 (ECRE 2022a).

---

[65] N.D. & N.T. v. Spain [GC], para. 201, App. Nos. 8675/15 and 8697/15 (Eur. Ct. H.R. 13 February 2020).

[66] A.A. & Others v. North Macedonia, para. 114, App. No. 55798/16 (Eur. Ct. H.R. 5 April 2022). See also id., para. 121 ("There is nothing in the case file to suggest that potential asylum-seekers were in any way prevented from approaching the legitimate border crossing points and lodging an asylum claim... or that the applicants attempted to claim asylum at the border crossing and were returned. The applicants in the present case did not even allege that they had ever tried to enter Macedonian territory by legal means.").

[67] See, e.g., Lang and Nagy (2021, p. 457) ("The judgment in N.D. and N.T.... sends a signal to EU member states that under certain conditions it is not illegal... to collectively push back third-country nationals who try to enter the EU, without individually assessing their status and knowing whether they are economic migrants or refugees."); Alonso Sanz (2021, p. 339) ("conceptual confusion" resulting in "a step backwards in the standard of protection of the rights of aliens against expulsion"); Ciliberto (2021, p. 220) (concluding that court's judgment "nurtures doubts on the scope of the prohibition of collective expulsion *vis-à-vis* interceptions of migrants on the seas, alongside the risk of having a (negative) impact on the effective and practical protection of the safeguards underpinning the principle of *non-refoulement*"); Thym (2020, p. 576) ("It is a general feature of the *ND and NT* judgment that the reasoning becomes nebulous when one scratches the surface of a seemingly clear-cut outcome.").

[68] Joint Dissenting Opinion of Judges Lemmens, Keller & Schembri Orland, para. 7, Asady & Others v. Slovakia, App. No. 24917/15 (Eur. Ct. H.R. 24 March 2020).

[69] M.H. & Others v. Croatia, App. Nos. 15670/18, 43115/18 (Eur. Ct. H.R. 18 November 2021); Shahzad v. Hungary, App. No. 12625/17 (Eur. Ct. H.R. July 8, 2021); D.A. & Others v. Poland, App. No. 51246/17 (Eur. Ct. H.R. 8 July 2021).

[70] Affaire Safi et Autres c. Grèce, Requête No. 5418/15 (Eur. Ct. H.R. 7 July 2022). See also Yaeger-Malkin (2022).

The Court of Justice of the European Union (CJEU) offers another important avenue to challenge pushbacks and other abusive limitations on access to asylum by E.U. member states.[71] The court has found, for example, that the Hungarian law allowing pushbacks of asylum seekers to Serbia was in breach of E.U. law.[72] It has also found that Lithuanian legislation preventing irregular migrants from applying for asylum and allowing mass detention during periods of "migrant influx" violated E.U. law.[73]

And in the context of transfers from one E.U. country to another under the Dublin III Regulation, which generally requires adults to seek asylum in the first E.U. country they reach, the CJEU has required states to assess whether disruptions in specialized care "would result in a real and proven risk of a significant and permanent deterioration in the state of health of the person concerned".[74] In line with the European Court of Human Rights, the CJEU has also precluded transfers to E.U. member states whose systematically deficient asylum procedures and inadequate reception conditions raise risks of subjecting people to inhuman or degrading treatment.[75]

In particular, the CJEU role in infringement proceedings against E.U. member states shows potential in situations in which states restrict access to asylum. The European Commission initiates infringement proceedings to secure compliance with E.U. law; if the initial stages of an infringement proceeding do not resolve the situation, the commission may refer a member state to the CJEU.[76] The commission has shown increasing willingness to make referrals for systematic rule-of-law shortcomings (see Komanovics (2022); Favi (2022); Bonelli (2022); Schmidt and Bogdanowicz (2021)), including undue restrictions on access to asylum.[77]

Domestic litigation has also yielded heartening results.

In 2020, Slovenia's Supreme Court upheld an administrative court's ruling that Slovenian authorities had violated a Cameroonian man's rights to seek asylum and to protection from refoulement when they detained him for two days, disregarded his repeated requests for asylum, and then returned him to Croatia, which then pushed him on to Bosnia.[78] The lower court had observed that before effecting returns under its readmission agreement, Slovenia has a proactive duty to verify whether Croatia respects the principle of nonrefoulement.[79]

Courts in Austria, Italy, and Switzerland have ruled that those countries' readmission agreements with neighboring states do not adequately safeguard against the risk of chain refoulement (from, for instance, Italy to Slovenia, Slovenia to Croatia, and then Croatia

---

71    As Maja Łysienia has observed, "In principle, the Luxembourg Court cannot be directly approached by individuals, including asylum seekers. It is not competent to decide that the fundamental rights of the concerned person were breached, grant just satisfaction and order specific measures that put an end to the situation that gave rise to a violation in the individual case. Those tasks are left to domestic courts and tribunals. However, the CJ may significantly affect decisions given on a national level by providing domestic authorities with the binding interpretation of the EU law." Łysienia (2022, p. 4). For an assessment of impediments to access by asylum seekers, see Łysienia (2022, pp. 94–95).

72    Case C-808/18 [GC], Eur. Comm'n v. Hungary, ECLI:EU:C:2020:1029 (CJEU 17 December 2020).

73    Affaire C-72/22 PPU, M.A. v. Valstybès sienos apsaugos tarnyba, ECLI:EU:C:2022:505 (CJEU 30 June 2022). See also ECRE (2022b).

74    Case C-578/16 PPU, C.K., H.F. & A.S. v. Republika Slovenija, ECLI:EU:C:2017:127 (CJEU 16 February 2017).

75    Compare Joined Cases C-411/10 & C-493/10, N.S. v. Secretary of State for the Home Dept. & M.E. & Others v. Refugee Applications Comm'r and Minister for Justice, Equality and Law Reform, ECLI:EU:C:2011:865 (CJEU 21 December 2011), with M.S.S. v. Belgium & Greece [GC], 2011-I Eur. Ct. H.R. 255; Tarakhel v. Switzerland [GC], 2014-VI Eur. Ct. H.R. 195. For a detailed discussion of *M.S.S.* and reflections on the extent to which the European Court of Human Rights has retreated from *M.S.S.* in subsequent destitution cases, see Dembour (2015, pp. 402–40, 455–56).

76    See Eur. Comm'n, Infringements: Frequently Asked Questions (17 January 2012), https://ec.europa.eu/commission/presscorner/detail/en/MEMO_12_12 (accessed on 14 February 2023).

77    See Case C-821/19 [GC], Eur. Comm'n v. Hungary, ECLI:EU:C:2021:930 (CJEU 16 November 2021); Case C-808/18 [GC], *supra* note 72.

78    Judgment and Order I Up 21/2020 (Slovn.), discussed in PIC (2022, p. 27).

79    Judgment and Order IU 1490/2019-92, paras. 69-72 (Admin. Ct. June 6, 2020) (Slovn.) (on file with author). See also Kramberger (2020); Bozic (2020).

to Bosnia and Herzegovina).[80] The Serbian Constitutional Court held that border police violated the Serbian constitutional prohibitions of expulsion and ill-treatment when they deported a group of Afghans to Bulgaria under the Serbia–E.U. readmission agreement.[81] Polish courts have found that border guards acted unlawfully by detaining and then pushing people to Belarus while disregarding their requests to apply for asylum.[82] And in Spain, courts have halted the returns of children from Ceuta to Morocco (Sánchez et al. 2021; Testa and Sánchez 2021).

Domestic court rulings have also issued favorable rulings in challenges to legislation that restricted access to asylum. For instance, after Slovenia amended its Aliens Act in 2017 to require police to disregard requests for asylum by people who had entered the country irregularly "if in the neighbouring European Union Member State from which the alien entered, there are no systemic deficiencies in relation to asylum procedure and reception conditions of asylum seekers which could cause danger of torture, inhuman and degrading treatment,"[83] the Constitutional Court held that the amendments violated the Slovenian constitutional prohibition of torture.[84]

Even the prospect of defending a legal challenge can prompt positive outcomes, at least for a time. In the United Kingdom, Prime Minister Boris Johnson's government attempted for months to allow pushbacks of small boats seeking to cross the Channel irregularly, despite warnings from the U.K. Parliament's Joint Committee on Human Rights that the proposal was incompatible with the United Kingdom's human rights obligations[85] and reports that the U.K. Border Force might strike or take legal action to oppose pushbacks (Slawson 2022; Syal 2021). When the Border Force union and three other groups challenged the pushback proposal in court, partly because it lacked explicit legislative authority, the Home Secretary withdrew it days before the case was scheduled to be heard (Syal 2022). That said, the subsequent enactment of the Nationality and Borders Act has provided an explicit legislative basis for authorities to "divert" boats, opening the door to pushbacks.[86]

As with the decisions of the European Court of Human Rights and the CJEU, compliance with domestic court rulings may be limited. For instance, although the Slovenian courts ordered the government to accept an asylum application from the Cameroonian man who had been pushed back to Bosnia and Herzegovina, the Ministry of the Interior did not

[80] Landesverwaltungsgericht Steiermark [LVwG] [Regional Administrative Court, Styria] 1 July 2021 docket No. 20.3-2725/2020-86 (Austria), http://asyl.at/files/514/3_000686_jv_sig_xx.pdf (accessed on 14 February 2023); Tribunale ordinario di Roma [ordinary court of first instance], 18 gennaio 2021, n. R.G. 56420/2020 (It.), https://www.asgi.it/wp-content/uploads/2021/01/Tribunale-Roma_RG-564202020.pdf (accessed on 14 February 2023); Tribunal administratif fédéral [TAF] [Federal Administrative Court] 6 January 2022 (Switz.), https://asile.ch/wp-content/uploads/2022/01/F-5675_2021.pdf (accessed on 14 February 2023). See also Vladisavljevic (2021); Pas de renvoi vers la Croatie sans examen approfondi du cas (2022).

[81] Case No. Už-1823/2017, 20 January 2021 (Const. Ct.) (Serb.), https://www.asylumlawdatabase.eu/sites/default/files/aldfiles/УЖ%201823-17.pdf (accessed on 14 February 2023). See also Raičević (2021).

[82] Action challenging detention by Narewka Border Guard post, Sygn. akt VII Kp 203/21 (Bielsko Podlaskie Dist. Ct. 28 March 2022) (Pol.), https://interwencjaprawna.pl/wp-content/uploads/2021/01/postanowienie-ws.-zatrzymania_VII_Kp_203_21.-zanonimizowane.pdf (accessed on 14 February 2023); Action challenging Border Guard Chief Commandant Decision No. r KG-CU-IV-2.4224.118.2021, Sygn. akt IV SA/Wa 615/22 (Voivodship Admin. Ct. Warsaw 20 May 2022) (Pol.) (on file with author). See also Stowarzyszenie Interwencji Prawnej (2022); L. Gall (2022).

[83] Aliens Act art. 10b(2) (Slovn.), quoted in Zagorc (2017).

[84] Decision U-I-59/17 (Constitutional Ct. 18 September 2019) (Slovn.), https://www.us-rs.si/decision/?lang=en&q=U-I-59%2F17-27&id=113724 (accessed on 14 February 2023).

[85] House of Commons & House of Lords, Joint Comm. on Human Rights, Legislative Scrutiny: Nationality and Borders Bill (Part 3)–Immigration Offences and Enforcement, Ninth Report of Session 2021-22 (1 December 2021), https://committees.parliament.uk/publications/8021/documents/83303/default/ (accessed on 14 February 2023).

[86] See Nationality and Borders Act 2022, c. 36, § 45 & sched. 7, § 10 (U.K.), https://www.legislation.gov.uk/ukpga/2022/36/contents/enacted (accessed on 14 February 2023). The act now allows U.K. authorities to require vessels they stop and board "to be taken to any place (on land or on water) in the United Kingdom *or elsewhere* and detained there" or to "*require the ship to leave United Kingdom waters.*" *Id.* sched. 7, § 10, Part 1A, sec. B1(2) (c) & (d) (amending Immigration Act 1971, c. 77, sched. 4A (enforcement powers in relation to ships), https://www.legislation.gov.uk/ukpga/1971/77/contents) (accessed on 14 February 2023).

issue him a temporary visa or otherwise facilitate his return from Bosnia and Herzegovina. The man was able to apply for asylum only after he transited Croatia irregularly and on his own initiative.[87]

Similarly, the Slovenian government attempted to circumvent the Slovenian Constitutional Court's decision to annul the 2017 amendments to the Aliens Act by enacting new legislation that allows border closures and restricted access to asylum.[88] These new measures are themselves before the Constitutional Court for review (Stranke KUL 2022).

Nonetheless, cases like the ones discussed above are powerful repudiations of abusive practices.

## 6. Conclusions

In Europe and elsewhere, refugee protection is at something of an inflection point. It is most obviously and directly under threat in states whose governments regularly resort to xenophobic and other hateful rhetoric and forthrightly attack the rule of law and democratic institutions. More generally and more insidiously, it is undermined by measures that "reshape and reinterpret" international obligations, steps that in the view of Jamal Barnes and Samuel M. Makinda "abid[e] by the letter of the law, while at the same time violat[e] its spirit".[89]

It is more accurate, however, to describe these measures as casuistic legal loopholes that avoid, rather than permit, compliance with international standards. Bilateral readmission agreements, exceptional restrictions that can easily become the norm, transit zones and similar legal fictions, the externalization of border controls, and other devices are all too readily deployed in ways that defeat the purpose of international protection and result in real harm. The principle of good faith compliance with international obligations[90] does not allow for legal sophistry.

Pushbacks take matters to an even greater extreme. They are often clandestine operations, unacknowledged and frequently denied by the authorities that carry them out. When these practices come to light or, as in Hungary and the United States, are affirmatively and publicly owned, authorities usually shield the operations themselves from meaningful public scrutiny. And some states that engage in pushbacks argue that they are working in a context where human rights norms somehow do not apply.

These persistent practices exact an enormous toll on everyone, regardless of age, as Maša Vukčević Marković, Aleksandra Bobić, and Marko Živanović found when they worked with refugees residing in Serbia (Marković et al. 2023). For children, the stakes can be particularly high. As Ryan Matlow, Alan Shapiro, and Ewen Wang have noted in the context of children travelling from Central America through Mexico to the United States, children's migration journeys are "rife with experiences of significant and severe stress, adversity, and trauma" (Matlow et al. 2023), putting them at heightened risk of toxic stress.[91]

---

[87] Interview in Ljubljana, Slovenia, 24 November 2021.

[88] PIC (2022); Republika Slovenija, Varuh Človekovih Pravic (Human Rights Ombudsman), *Varuh Evropsko Komisijo seznanil s svojimi pogledi na novelirano tujsko zakonodajo* [The Ombudsman informs the European Commission of his views on amended foreign legislation], 16 August 2021, https://www.varuh-rs.si/sl/sporocila-za-javnost/novica/varuh-evropsko-komisijo-seznanil-s-svojimi-pogledi-na-novelirano-tujsko-zakonodajo/ (accessed on 14 February 2023).

[89] Barnes and Makinda (2021). In a similar vein, Barnes has said elsewhere, "Policies such as pushbacks and pullbacks, expulsions, and extraterritorial and arbitrary detention, among other deterrence policies, have been implemented not by avoiding international law but using it in a way that adheres to the letter but not the spirit of the law" Barnes (2022, p. 446).

[90] Vienna Convention on the Law of Treaties pmbl. & art. 26, 23 May 1969, 1155 U.N.T.S. 331. See also Lukashuk (1989, p. 515) ("[S]tates are under an obligation to refrain both from acts defeating the object and purpose of a rule and from any other acts preventing its implementation").

[91] Research by Jack Shonkoff and Andrew S. Garner has found that toxic stress in children can change brain structure and function and "lead to potentially permanent changes in learning (linguistic, cognitive, and social-emotional skills), behavior (adaptive versus maladaptive responses to future adversity), and physiology (a hyperresponsive or chronically activated stress response)" Shonkoff and Garner (2012, p. e243).

Even so, and despite robust international norms, it is easy to take the successful court challenges to pushback practices in domestic and regional courts as aberrations. But the number of positive judgments is encouraging, as is the range of jurisdictions.

To be sure, compliance with positive domestic court decisions has been uneven, and in other cases, tribunals have deferred unduly to the government. The European Court of Human Rights has signaled that it may be willing to overlook some critical safeguards for those who enter the European Union irregularly. The European Commission may be reluctant to pursue infringement proceedings against countries other than Hungary that disregard refugee protections. The commission's efforts to establish border monitoring mechanisms have not yet lived up to their potential. The European Union as a whole has not resolved an inherent unfairness in its asylum framework which unduly burdens countries at the union's external borders.

And despite the commitment in the Global Compact on Migration to expanded pathways for safe and legal migration channels,[92] the main destination countries have not yet lived up to this and other undertakings in this compact and the parallel Global Compact on Refugees.

It is worth noting that nearly 60 percent of forcibly displaced people worldwide are internally displaced—they have not left their countries of origin or residence. Of those who do cross an international border in search of safety, more than 80 percent are hosted in low- and middle-income countries, according to UNHCR.[93] Efforts by Australia, European Union member states, the United Kingdom, and the United States to undermine international protection makes it harder for those countries to call on countries in the global south not to follow suit.

Even so, and despite the widespread use of pushbacks and other abusive practices, the commitments set forth in the compacts, the willingness of some domestic courts to check the worst governmental overreach, and regional avenues for some measure of accountability are hopeful signs. For advocates, the challenge is how to make the most of these and other opportunities, however limited.

For states and the European Union, the tasks are to ensure good-faith compliance with international obligations (including by ensuring that border monitoring mechanisms are independent and effective), fulfil the promises made in the global compacts, and resist the lure of abuse in the service of xenophobic symbolism.

**Funding:** This research received no external funding.

**Institutional Review Board Statement:** Not applicable.

**Data Availability Statement:** No new data were created or analyzed in this study. Data sharing is not applicable to this article.

**Acknowledgments:** The author would like to thank Warren Binford, Eva Cossé, Bill Frelick, Emilie McDonald, Sophie McNeill, Elaine Pearson, Aisling Reidy, Ari Sawyer, Bede Sheppard, Jude Sunderland, Giulia Tranchia, Bill Van Esveld, and Ben Ward for their comments on the drafts of this article.

**Conflicts of Interest:** The author declares no conflict of interest.

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
