# Peer review of "The Persistent, Pernicious Use of Pushbacks against Children and Adults in Search of Safety"

_laws, 2022_

Round 1

Reviewer 1 Report

Perhaps consider whether the conclusion can be made a bit more robust-- there are compelling and well-supported arguments throughout that seem to be given short shrift in the conclusion. The conclusion seems to tail off, providing a somewhat attenuated discussion to summarize the article-- this does not seem to do justice to the strength of the article and its arguments.

Author Response

Thanks for this helpful feedback. In response to these and other reviewers' comments, I have added a paragraph on methods and sources to the introduction, updated the pushback numbers to reflect the latest data, and made more explicit reference to the children's rights impacts of pushbacks in the summary and the conclusion.

Reviewer 2 Report

There is no theoretical basis, so it is only a description. There is no reference to the Ukraine War. How can it be? 

Author Response

The article follows a standard legal analysis of presenting legal principles and statements of policy, contrasting actual practice (the "description" you refer to) against those principles, and then assessing efforts to secure compliance with those principles. 

In response to the comment on the Ukraine war, I have inserted mention of people fleeing Ukraine as a result of Russia's invasion and noted the contrast between the welcome given to Ukrainian refugees versus the abusive treatment frequently inflicted on others seeking safety.

In addition, responding to other reviewers' comments, I have added a paragraph on methods and sources to the introduction, updated the pushback numbers to reflect the latest data, and made more explicit reference to the children's rights impacts of pushbacks in the summary and the conclusion.

Reviewer 3 Report

The aim of the article is to document the increased practice of border pushback, with a specific focus on children.  The analysis is based on documents, reports and research describing the practices and implications of pushback in different regions and by various countries. 

The article is well structures, and the documentation persuasive. 

My main advice is to include a paragraph on data and methods, a clear research question and a presentation of the structure and argument in the first section. The stress on children and pushback could also be more explicit addressed here and also related to theories/research on children and migration control. And also how and when data/information on children will be included in the next sessions.

And then to include these research questions in the conclusion. Now, the main argument is unpacked and implicit.

To include the child perspective more systematic in the different section of the article - one option in section V - then is to include specific information on or reflect on how assessments are done related to children when referring to the case laws and court decisions of European Court of Human Rights, CJEU and on country level. 

In the conclusion it is also needed to address children - eg. related to the international obligations specified to migrant children. And maybe also related to the effort to establish border monitoring mechanisms. 

Author Response

Thank you for your helpful comments. In response to these and other reviewers' comments, I have added a paragraph on methods and sources to the introduction, updated the pushback numbers to reflect the latest data, inserted mention of people fleeing Ukraine as a result of Russia's invasion and noted the contrast between the welcome given to Ukrainian refugees versus the abusive treatment frequently inflicted on others seeking safety, and made more explicit reference to the children's rights impacts of pushbacks in the summary and the conclusion.